# Safety Performance of Healthcare Professionals: Validation and Use of the Adapted Workplace Health and Safety Instrument

**DOI:** 10.3390/ijerph18157816

**Published:** 2021-07-23

**Authors:** Lina Heier, Nikoloz Gambashidze, Judith Hammerschmidt, Donia Riouchi, Matthias Weigl, Andrew Neal, Andrea Icks, Peter Brossart, Franziska Geiser, Nicole Ernstmann

**Affiliations:** 1Institute for Patient Safety (IfPS), University Hospital Bonn, 53127 Bonn, Germany; judith.hammerschmidt@ukbonn.de (J.H.); do.riouchi@gmail.com (D.R.); matthias.weigl@ukbonn.de (M.W.); franziska.geiser@ukbonn.de (F.G.); nicole.ernstmann@ukbonn.de (N.E.); 2Center for Health Communication and Health Services Research (CHSR), Department for Psychosomatic Medicine and Psychotherapy, University Hospital Bonn, 53127 Bonn, Germany; 3School of Psychology, Faculty of Health and Behavioral Sciences, University of Queensland, Brisbane 4027, Australia; a.neal@psy.uq.edu.au; 4Institute for Health Services Research and Health Economics, Centre for Health and Society, Medical Faculty and University Hospital Düsseldorf, Heinrich-Heine-University Düsseldorf, 40225 Düsseldorf, Germany; Andrea.Icks@uni-duesseldorf.de; 5Department of Oncology, Hematology, Immuno-Oncology and Rheumatology, University Hospital Bonn, 53127 Bonn, Germany; peter.brossart@ukbonn.de; 6Department for Psychosomatic Medicine and Psychotherapy, University Hospital Bonn, 53127 Bonn, Germany

**Keywords:** patient safety, occupational safety, safety performance, healthcare professionals, acute care

## Abstract

Improving patient safety and reducing occupational accidents are two of the main challenges in healthcare. Instruments to measure safety performance and occupational safety are rare. This study aimed to prepare and validate a German version of the adapted workplace health and safety instrument to assess the safety performance of healthcare professionals. Overall, 168 healthcare professionals participated in this explorative cross-sectional study. The instrument consists of 16 items related to safety performance in four dimensions. We calculated mean values and standard deviations for each individual item and those of the four dimensions of the instrument. We evaluated internal consistency and construct validity, explored the dimensionality of the instrument through exploratory factor analysis, and tested how our data fit with the original model with confirmatory factor analysis. Among the participants, 73.8% were nurses and nurses in training, with the majority of the sample being female (71.9%) and younger than 30 (52.5%). Cronbach’s alpha for all four dimensions was >0.7. All items were loaded on factors according to the original theoretical model. Confirmatory factor analysis showed good model fit (normed χ²/df = 1.43 (≤2.5), root mean square error of approximation = 0.06 (≤0.07), goodness of fit index = 0.90 (>0.90), comparative fit index = 0.95 (≥0.90), and Tucker–Lewis index = 0.93 (>0.90). The German version of the instrument demonstrated acceptable properties and was a good fit to the original theoretical model, allowing measurement of healthcare professionals’ safety knowledge, motivation, compliance, and participation.

## 1. Introduction

Thousands of deaths and disabilities occur because of occupational accidents each year worldwide; approximately 318,000 of these deaths are due to accidents and two million are due to work-related diseases [1,2,3]. In Germany, 22.8% of all occupational accidents in 2019 were among healthcare professionals working in the public sector [4]. One in every ten patients is harmed while receiving acute care in hospitals [5]. Improving patient safety and reducing occupational accidents are two of the main challenges in healthcare worldwide [6]. A growing body of literature focusing on the impact of an organizational or safety climate, safety performance, and occupational safety, as well as occupational safety climate in healthcare, has emerged [7,8,9,10,11,12,13,14,15,16]. Studies explicitly focusing on safety performance in acute care hospitals are rare, although healthcare professionals play a leading role in improving and maintaining patient safety. Healthcare providers’ own behaviors, together with workplace safety, affect occupational and patient safety [17,18] with strong links to nontechnical skills including error awareness, teamwork, and decision-making [19,20].

According to Griffin and Neal, safety performance consists of safety compliance and safety participation [21]. The term safety compliance is used to describe the core activities that individuals need to carry out to maintain workplace safety. These behaviors include adhering to standard work procedures and wearing personal protective equipment. The term safety participation is used to describe behaviors that do not directly contribute to an individual’s personal safety but help develop an environment that supports safety [21,22]. With regard to the integrative model of workplace safety, safety performance is directly influenced by the proximal person-related factors of safety motivation and safety knowledge [23] (Figure 1). Safety motivation and safety knowledge are supposed to be influenced by safety climate, leadership, personality characteristics, and job attitudes [13,23]. Knowledge is a precondition for enacting safe behaviors (the know-how to perform safely), and motivation reflects a willingness to attempt to use safety behaviors [23,24]. The model of workplace safety of Griffin and Neal has been widely used and successfully implemented within occupational safety across different organizational contexts [23]. Kaleth and colleagues tested a Persian safety performance scale among Iranian employees of a petrochemical complex and reported a high internal consistency [25]. Similar results were reported by Braunger and colleagues, who tested the instrument as well as the model in Austria [26]. Gracia and colleagues underline their findings for the Spanish setting [27]. Toderi and colleagues studied the individual safety performance in Italy and their findings confirmed the adequacy of the original items, and they reported a satisfactory reliability as well as validity [28]. Scales that measure the constructs of safety knowledge, safety motivation, safety compliance, and safety participation were presented in the original study of Neal et al. [22]. This model and the workplace health and safety instrument used by Neal et al. are widely used and proven to be stable, with good psychometric properties and excellent internal consistency [22,23]. However, to our knowledge, it has not yet been used in healthcare settings in German-speaking countries.

This study aimed to prepare and validate a German version of the adapted workplace health and safety instrument to assess the safety performance of healthcare professionals in German hospitals.

## 2. Materials and Methods

### 2.1. Design

A stepwise procedure was established with translation, application, and testing for psychometric and factorial validity. The data for this study was collected as part of the research project “Safety Performance of Healthcare Professionals” (SPOHC), conducted in 2018–2020. The ethics committee of the University Hospital Bonn approved the study methodology (number: 075/19).

### 2.2. Study Instrument

The original English version of the workplace health and safety instrument (WHASI) published by Neal et al. was translated into German [22]. In this study, we followed the technique recommended by van de Vijver, Banville et al., and Vallerand [29,30,31]. A double translation/back-translation was conducted by three different professional translators. The translated items were then discussed and evaluated by three researchers in cooperation with one of the developers of the original scales to ensure that the meanings of the original items were retained. The German version of WHASI (WHASI-G), which is similar to the original instrument, consists of 16 items related to the safety behaviors of clinical personnel. Consistent with the English version, items can be grouped into four scales: Safety Motivation, Safety Knowledge, Safety Compliance, and Safety Participation. Participants’ agreement is measured using a five-point Likert scale (from 1 (strongly disagree) to 5 (strongly agree)). The German version was cognitively pretested with nursing staff from a long-term care facility in Germany (*n* = 12) [32]. This method aimed to identify the causes of errors in surveys [32]. Additional items captured the demographic characteristics of the study participants.

### 2.3. Setting, Procedure, and Sample

Using the WHASI-G, the data was collected in three hospitals and two nursing schools. First, the organizations’ top management was informed about the goals and methods of the study via email. Next, a member of the study team met the decision-makers to present the study and clarify the questions. After the top management agreed to participate, they suggested clinics and their teams. These units and teams were informed about the study by the study team via email. If the heads of the units/teams were interested in participating, they were asked for an appointment to present the objectives, data protection, contact persons, and next steps. After all the questions have been answered, all healthcare professionals of these teams were invited to participate in the study. We invited them via email and personal contact in team meetings. Reminders were sent out to facilitate engagement after two and four weeks. The questionnaire was preceded by an informed consent form that provided participants with detailed information about the study. Participation was completely voluntary and anonymous.

### 2.4. Analysis

We calculated the mean values and standard deviations for each individual item, as well as those of the four dimensions of the instrument [33]. The dimension scores were calculated by averaging corresponding items. The number (percentage) of missing answers for individual items was evaluated as an indication of acceptability. The floor and ceiling effects, measured as the proportion of the lowest and highest answer choices, respectively, were evaluated to check the instrument’s performance on the low and high construct levels.

To evaluate the psychometric properties of the WHASI-G, we analyzed internal consistency, construct, and factorial validity. The internal consistency of each dimension was evaluated using Cronbach’s alpha, with an expected alpha of >0.7. To evaluate construct validity, we studied Spearman’s correlations and expected low to moderate positive correlations between the four safety performance dimensions. We evaluated convergent validity by calculating the average variance extracted (AVE) for each dimension (AVE > 0.5). We estimated divergent validity using the Fornell-Larcker criterion. We calculated the square root of AVE (√AVE) and expected it to be larger compared to correlations between dimensions [34].

To evaluate if the data were suitable for factor analysis, we first calculated the Kaiser–Meyer–Olkin (KMO) and measure of sampling adequacy (MSA) (for both >0.7 desired and >0.9 perfect). Additionally, the significant *p*-value (<0.05) of Bartlett’s test of sampling adequacy would indicate that it is possible to extract more than one factor. Most of these analyses used the list-wise exclusion of missing cases; therefore, only complete cases were used for the analysis. To explore the factor structure based on our data, we conducted exploratory factor analysis (EFA) with maximum likelihood estimation. A scree plot and Eigenvalues of >1 guided the factor extraction. To aid in the interpretation of factors, we used varimax orthogonal pre-rotation followed by promax oblique rotation. Factor loadings of >0.35 were considered significant, and cross-loadings of <0.35 were considered acceptable [33].

CFA tests the fit of empirical data with the originally proposed model (i.e., four-factor structure). We used the following fit indices with corresponding benchmarks: normed χ² (χ²/df ≤ 2.5), root mean square error of approximation (RMSEA ≤ 0.07), the goodness of fit index (GFI > 0.90), comparative fit index (CFI ≥ 0.90), and Tucker–Lewis index (TLI > 0.90) [35]. We conducted CFA using the four-factor model proposed by Neal et al. and compared fit indices with the results of a one-factor solution [22].

## 3. Results

### 3.1. Study Sample and Descriptive Statistics

Thirteen departments from three hospitals and two nursing schools were included in the study. A total of 430 HCPs were invited to participate. The response rate was 39.1% (*n* = 168). After inspection, 8 cases were removed because all of the 16 WHASI-G items were missing. The remaining 160 cases were available for further analysis. Demographic characteristics of the study sample are provided in Table 1. Nurses and nurses in training comprised 73.8% of the participants, with the majority of the sample being female (71.9%) and without leadership functions (82.5%). About half of the participants were younger than 30 (52.5%).

All items had missing answers of <5%, ranging from 0% to 3.8%. Item 13 (“I promote the safety program within the organization”) demonstrated a floor effect (lowest score of >15%); all other items and all four dimensions demonstrated a ceiling effect (highest score of >15%). The mean scores for the four dimensions ranged from 3.5 for Safety Participation, to 4.83 for Safety Motivation. Descriptive statistics are presented in Table 2.

### 3.2. Analysis of Internal Consistency, Construct, and Factorial Validity

Of the 160 overall cases, 141 complete cases (88.1%) were available for further analysis. Table 3 shows the internal consistencies of the WHASI-G tool and the correlations between scales. The Cronbach’s alpha for all four dimensions was >0.7. Further analyses revealed that, if item 5 (“I believe that workplace health and safety is an important issue”) were removed, Cronbach’s alpha would increase from 0.72 to 0.78. The correlations between the WHASI-G’s four dimensions were all positive and in the small to moderate range, with all but one being significant at *p* < 0.01. The dimension Safety Compliance demonstrated adequate convergent validity (AVE = 0.54), while the other three dimensions had AVE < 0.5. All four dimensions demonstrated discriminant validity as √AVE of all dimensions were higher than correlations with other dimensions.

The KMO for the sample of 141 complete cases was 0.79, and the MSA for individual items were all >0.7, except for items 5 (0.54) and 8 (0.63), which are both from the scale Safety Motivation. Bartlett’s test of sampling adequacy was significant (*p* < 0.001), indicating that more than one factor could be extracted.

Based on Keiser’s criterion (Eigenvalues > 1) and the evaluation of the scree plot, four factors were extracted in EFA. Table 4 presents the final, rotated solution and the items’ factor loadings. All WHASI-G items were loaded on factors according to the original theoretical model. None of the items had significant cross-loading. The item with the lowest factor loading was item 5 from the scale Safety Motivation (see Table 4).

The sample size did not allow for split-half cross-validation, conducting EFA and CFA in separate halves. Consequently, we aimed to use CFA to test the fit of the originally proposed model (i.e., four-factor model) with the data. CFA showed a good fit of the original four-factor model to our empirical data: normed χ²/df = 1.43 (≤2.5), RMSEA = 0.06 (≤0.07), GFI = 0.90 (>0.90), CFI = 0.95 (≥0.90), and TLI = 0.93 (>0.90). In comparison, all of the fit indices of the one-factor solution were below the set criteria: χ²/df = 4.1 (≤2.5), RMSEA = 0.15 (≤0.07), GFI = 0.71 (>0.90), CFI = 0.59 (≥0.90), and TLI = 0.53 (>0.90).

## 4. Discussion

In this study, we aimed to prepare the German version of WHASI and evaluate its psychometric properties. We used data from three German hospitals to validate the WHASI-G for use in German-speaking healthcare facilities.

In our study, the WHASI-G demonstrated acceptable properties in most of the analyses and was a good fit for the original theoretical model. Furthermore, there were very few missing answers, indicating that participants were able to respond to all questionnaire items. Only one item demonstrated floor effect, and all the other items had a significant ceiling effect, meaning that the instrument may not be able to detect differences on the high end of the measurement scale. Item 5 of the scale Safety Motivation (“I believe that workplace health and safety is an important issue”) seemed uncontroversial, and consequently, most of the study participants completely agreed with it (93.1%). The highly limited variance of this item was reflected in a lower correlation with the other items and poorer performance. Still, it demonstrated an acceptable fit with the general model; therefore, we would not recommend removing this item but instead using the instrument in the original form. The three out of four dimensions of the instrument demonstrated limited convergent validity. At the same time, all four dimensions had adequate discriminant validity indicating, that they measure distinctly different constructs.

The results of the German scales support the adequacy of the original version as well as those found in previous studies examining the adapted version of the WHSI. The four-factor model proposed by Neal et al. can be suggested to properly measure determinants and components of safety performance [22]. The correlations between the WHASI-G’s four dimensions were all positive, with all but one being significant at *p* < 0.01, which is similar to the findings of Toderi and colleagues, who tested the four-factor model in the Italian context [28]. In the present study, the Cronbach’s alpha for all four dimensions was >0.7, which is similar to the findings of Kwon and Kim as well as the original study of Neal et al. [22,36]. The Spanish version reported a Cronbach’s α value of 0.88 and 0.86 of safety compliance and safety participation, which also underlines the good psychometric properties with a high internal consistency [37]. CFA produced an excellent fit for the 4-factors model, e.g., with a RMSEA = 0.06 (≤0.07), and CFI = 0.95 (≥0.90). In comparison, the Italian version showed an RMSEA = 0.04 and CFI = 0.99 [28].The adapted integrative model of workplace safety is based on Griffin and Neal’s model of safety performance, which itself is built upon the theory of performance [21]. In summary, the adapted model is characterized by two main concepts: on the one hand, the distinction between safety compliance and safety participation as part of the construct of safety performance and, on the other, the distinction of proximal person-related factors of safety knowledge and safety motivation [23]. Concerning the descriptive results among German healthcare professionals working in acute care, our data showed lower safety participation scores compared with the safety compliance scores. These results are similar to other studies that have used the same instrument [26,28]. The original instrument developed and published by Neal et al. distinguished between safety activities that are part of the job (safety compliance) and activities that support the broader environmental context (safety participation) [22]. A meta-analysis conducted by Christian et al. showed that safety participation and safety compliance may be influenced by aspects of safety climate, such as workers’ attitudes and beliefs regarding safety [23]. Similarly, longitudinal studies by Neal and Griffin suggested that an employee’s safety motivation and safety participation can improve if they believe in the importance of safety [24,38]. Manapragda et al. also found that, when nurses believe that management is committed to safety, promotes open communication, and supports safety systems, they are more likely to stick to safety practices and promote the safety agenda of their workplace [39]. In our study, no safety climate aspects were collected. There is a need for further research to investigate the extent to which the aspects of safety climate may explain the difference between safety participation and safety compliance scores.

### 4.1. Limitations

The results of this investigation should be interpreted in light of several limitations. The main limitation of our study is the convenience of our sampling approach and the proportions of the surveyed professions. We used a systematic, stepwise approach to explore the applicability of the newly developed WHASI-G. Yet, we acknowledge that our sample size was limited and that its composition may limit the external validity of our results; i.e., the majority of the participants were registered nurses and nurses in training. It must also be noted, that the topic of safety performance in acute care can be strongly influenced by social desirability bias. In addition, there is a potential risk of selection bias due to the recruitment strategy of this study. Not all clinics and teams of the participating hospitals were contacted by the study team, but only those suggested by management. It remains unclear whether management has followed certain characteristics of the proposed units and teams. Future studies should thus test the WHASI-G among larger, more professionally diverse samples to evaluate its applicability in multi-professional teams across different care domains, e.g., acute vs. non-acute and inpatient vs. outpatient care facilities. Due to the sensitive nature of safety performance, participation was completely anonymous, and consequently evaluating test-retest validity was not part of the study. The results were obtained from professionals working in hospitals in Germany and thus may reflect safety measures and standards specifically for this national healthcare system. Future investigations into the content and criterial validity of the WHASI-G should be undertaken, such as the concurrent measurement of safety performance with an alternative tool or the incorporation of measures of the workplace and patient safety. In our study, we used an adapted model that did not include measuring other determinants that influence safety, such as safety climate or leadership aspects. Future studies should further examine the association between these constructs and their effect on external measurements, such as patient-related outcomes.

### 4.2. Practical Implications

The WHASI-G, tested and validated in this study, provided a way to measure registered nurses’, nursing students’, and physicians’ safety performance, safety knowledge, and safety motivation in a clinical setting. This 16 items instrument is easy to use and can be implemented to identify gaps in these particular dimensions. With regard to the recently published World Health Organization (WHO) global patient safety action plan, which provides concrete actions to be taken by health care facilities [6], hospitals and their clinical teams can use WHASI-G to measure their safety performance, to develop interventions to improve patient safety and/or occupational safety.

In academia, the instrument enables researchers to study the role of safety performance, safety knowledge, and safety motivation in high-reliability organizations. It remains outstanding, which role sociodemographic and/or organizational aspects have and how these factors (e.g., profession, gender) predict the safety performance of healthcare professionals. These aspects should be addressed in further studies to get a more comprehensive understanding of safety performance and its related factors.

## 5. Conclusions

Drawing upon a stepwise procedure, we successfully adapted and tested the German version of the workplace health and safety instrument (WHASI-G). It can be used for the evaluation of the safety performance of clinical teams in German hospitals. The WHASI-G demonstrated good psychometric properties and showed good factorial validity consistent with the original version proposed by Neal et al. [22]. The WHASI-G is a reliable, easy-to-use instrument for measuring and monitoring the safety-related knowledge, motivation, compliance, and participation of healthcare professionals in clinical care environments. In the future, it can be applied by management, administrative, or clinical personnel to develop and evaluate training or safety-related interventions in healthcare. This work makes an important practical contribution, as the availability of a validated instrument for German-speaking countries to measure safety performance supports both the development of safety research and clinical risk management strategies.

## Figures and Tables

**Figure 1 ijerph-18-07816-f001:**
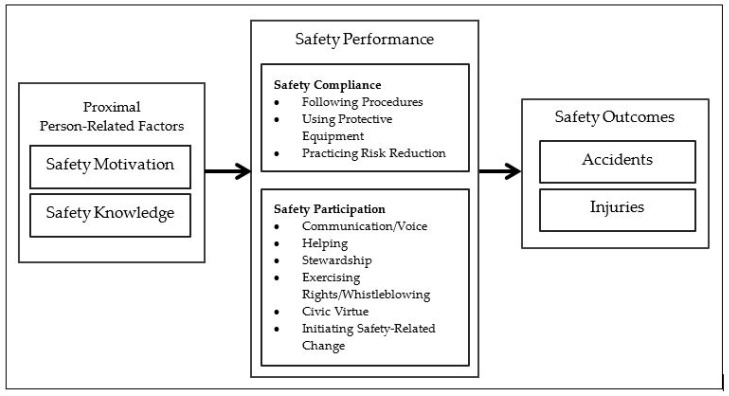
Adapted integrative model of workplace safety [23]. Adapted with permission from Christian, M.S.; Bradley, J.C.; Wallace, J.C.; Burke, M.J. (2009). 2021 Michael Christian.

**Table 1 ijerph-18-07816-t001:** Characteristics of the study sample.

	*n*	%
**Total sample**	160	100%
**Profession**		
Physician	23	14.40%
Nurse	62	38.80%
Nurse in training	56	35.00%
Other	18	11.30%
Missing	1	0.60%
**Gender**		
Female	115	71.90%
Male	45	28.10%
**Leadership functions**		
Yes	24	15.00%
No	132	82.50%
Missing	4	2.50%
**Tenure in the profession**		
<3 months	2	1.30%
3–12 months	1	0.60%
1–5 years	76	47.50%
>5 years	79	49.40%
Missing	2	1.30%
**Tenure in clinic**		
<3 months	4	2.50%
3–12 months	6	3.80%
1–5 years	87	54.40%
>5 years	62	38.80%
Missing	1	0.60%
**Age**		
<30	84	52.50%
31–40	29	18.10%
41–50	23	14.40%
>50	23	14.40%
Missing	1	0.60%

Note: Demografic variables are presented in bold.

**Table 2 ijerph-18-07816-t002:** Descriptive statistics of the WHASI-G items and scales.

Dimensions/Items	Missing	Floor Effect	Ceiling Effect	Mean Score	SD
**Safety Knowledge**	**0.0%**	**0.0%**	**42.5%**	**4.19**	**0.63**
1. I know how to perform my job in a safe manner.	0.0%	0.0%	45.0%	4.35	0.68
2. I know how to use safety equipment and standard work procedures.	1.9%	0.6%	41.9%	4.15	0.89
3. I know how to maintain or improve workplace health and safety.	3.1%	0.0%	31.9%	4.08	0.80
4. I know how to reduce the risk of accidents and incidents in the workplace.	2.5%	0.0%	36.3%	4.19	0.77
**Safety Motivation**	**0.0%**	**0.0%**	**91.3%**	**4.83**	**0.30**
5. I believe that workplace health and safety is an important issue.	0.0%	0.0%	93.1%	4.90	0.42
6. I feel that it is worthwhile to put in the effort to maintain or improve my personal safety.	1.9%	0.0%	78.1%	4.77	0.49
7. I feel that it is important to maintain safety at all times.	0.6%	0.0%	78.8%	4.77	0.46
8. I believe that it is important to reduce the risk of accidents and incidents in the workplace.	0.6%	0.0%	88.8%	4.89	0.34
**Safety Compliance**	**1.3%**	**0.0%**	**38.8%**	**4.18**	**0.64**
9. I carry out my work in a safe manner.	1.3%	0.0%	36.9%	4.22	0.74
10. I use all the necessary safety equipment to do my job.	3.1%	1.3%	44.4%	4.21	0.90
11. I use the correct safety procedures for carrying out my job.	2.5%	0.6%	33.1%	4.21	0.72
12. I ensure the highest levels of safety when I carry out my job.	2.5%	0.6%	29.4%	4.10	0.76
**Safety Participation**	**0.6%**	**1.3%**	**17.5%**	**3.50**	**0.87**
13. I promote the safety program within the organization.	3.8%	18.8%	8.1%	2.78	1.24
14. I put in extra effort to improve the safety of the workplace.	1.3%	3.1%	19.4%	3.53	1.04
15. I help my coworkers when they are working under risky or hazardous conditions.	1.3%	1.3%	52.5%	4.41	0.77
16. I voluntarily carry out tasks or activities that help to improve workplace safety.	2.5%	11.3%	18.1%	3.28	1.26

Note: Mean scores represent low (1) to high (5) agreement. SD: standard deviation. Dimensions, presented in bold, according to the original instrument of Neal et al. [22].

**Table 3 ijerph-18-07816-t003:** Internal consistency (Cronbach’s alpha), convergent and discriminant validity, and construct validity (Spearman’s rho).

	α	AVE	√AVE	Spearman’s Correlation		
				SK	SM	SC
SK—Safety Knowledge	0.80	0.46	0.68			
SM—Safety Motivation	0.72	0.45	0.67	0.24 *		
SC—Safety Compliance	0.84	0.54	0.73	0.54 *	0.31 *	
SP—Safety Participation	0.76	0.44	0.66	0.33 *	0.14 +	0.43 *

Note: α—Standardized alpha; AVE—average variance extracted; √AVE—square root of average variance extracted; * *p* < 0.01; + *p* = 0.09; SK—Safety Knowledge, SM—Safety Motivation, SC—Safety Compliance.

**Table 4 ijerph-18-07816-t004:** Results of exploratory factor analyses for the WHASI-G (rotated solution).

Items	Factor 1	Factor 2	Factor 3	Factor 4
**Safety Knowledge**				
1. I know how to perform my job in a safe manner.	**0.569**	−0.016	0.190	0.062
2. I know how to use safety equipment and standard work procedures.	**0.576**	−0.001	0.044	0.148
3. I know how to maintain or improve workplace health and safety.	**0.805**	0.090	−0.042	−0.078
4. I know how to reduce the risk of accidents and incidents in the workplace.	**0.744**	0.059	−0.043	−0.013
**Safety Motivation**				
5. I believe that workplace health and safety is an important issue.	0.121	**0.370**	−0.051	−0.026
6. I feel that it is worthwhile to put in the effort to maintain or improve my personal safety.	0.007	**0.630**	0.118	−0.066
7. I feel that it is important to maintain safety at all times.	0.136	**0.567**	0.064	−0.067
8. I believe that it is important to reduce the risk of accidents and incidents in the workplace.	−0.069	**0.968**	−0.054	0.077
**Safety Compliance**				
9. I carry out my work in a safe manner.	0.128	0.065	**0.635**	0.034
10. I use all the necessary safety equipment to do my job.	−0.063	0.062	**0.599**	0.090
11. I use the correct safety procedures for carrying out my job.	0.035	−0.112	**0.978**	−0.099
12. I ensure the highest levels of safety when I carry out my job.	−0.009	0.101	**0.660**	0.113
**Safety Participation**				
13. I promote the safety program within the organization.	0.057	−0.036	0.082	**0.696**
14. I put in extra effort to improve the safety of the workplace.	0.076	−0.027	−0.054	**0.709**
15. I help my coworkers when they are working under risky or hazardous conditions.	−0.155	0.130	0.131	**0.545**
16. I voluntarily carry out tasks or activities that help to improve workplace safety.	0.056	−0.100	−0.029	**0.677**

Note: In bold are loadings > 0.35. Dimensions, presented in bold, according to the original instrument of Neal et al. [22].

## Data Availability

The data presented in this study are available on request from the corresponding authors.

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
