# Peer review of "Safety Performance of Healthcare Professionals: Validation and Use of the Adapted Workplace Health and Safety Instrument"

_ijerph, 2021, doi:10.3390/ijerph18157816_

Round 1

Reviewer 1 Report

This study prepared and validated a German version of the adapted workplace health and safety instrument in order to assess safety performance of healthcare professionals. Some comments for the authors to improve the quality of the manuscript are as follows.

  1. The sampling method is missing. How to recruit the participants?
  2. It is better to have one sample for EFA to explore the factor structure of the WHASI-G and another sample for CFA to confirm the result of EFA.
  3. Test-retest reliability and Convergent and discriminant validity of the measurement should be investigated because they are also important in scale validation. The authors may refer to the work “Quantification of risk perception: Development and validation of the construction worker risk perception (CoWoRP) scale”The result of CFA is missing, including factor loading and average variance extracted.
  4. The authors should use the demographic data to predict the safety knowledge, safety motivation, safety compliance, and safety participation using linear regression.
  5. The results of this study should be compared with previous studies and discussed accordingly.
  6. There should be a section to discuss the practical implications of the results.

Reviewer 2 Report

This is a well written manuscript. While I am not an expert in the statistical methods used, it is clear to me what the four dimensions from the original model were and how and with whom they were tested. The discussion and conclusions were reasonable and I agree with the authors that this should be replicated with other professions and in the context of some of the other factors that influence knowledge and motivation. While generally clear and well-written, some English language editing is needed. There are some challenges with case agreement, tense, the oxford comma, and a few sentences that need reworking. Overall, this is will written and I commend the authors for their work.

In addition, I will share the attached that details the sentences that I found unclear and in need of revision.

Thank you for the opportunity to review.

Reviewer 3 Report

The authors have conducted an interesting research on assessing the safety performance of healthcare professionals. Some improvements are required before it can be considered for publication. My comments are as follows.

  • In the abstract, add a sentence related to the conclusions/ implications of the research.
  • If f Fig 1 has been taken from a reference, please get permissions, if needed.
  • You may mention the ethics approval number at the end of the manuscript, probably in the acknowledgment.
  • The authors should justify the
  • Line 123, reference is required.
  • In Table 2, the authors have listed some dimensions which is the basis for the analysis. Where did you get them from? Have you adopted these dimensions? If yes, you need to justify why no changes have been required for this adoption. In other words, you need to elaborate regarding the suitability of using such dimensions in the context of your research.
  • In the discussion section, please avoid citing unnecessary references. These citations should be made in the introduction/ literature section.
  • In conclusion, please highlight the academic/practical implications with more details.  

Round 2

Reviewer 1 Report

The authors failed to address my critical comments even they made a little effort. First, they did not mention the sampling methods used in the study to avoid possible bias and collect useful data. In this study, apart from 58 nursing students, only 102 professionals were involved. 62 were nurses, which may lead the results to be biased. Second, they reject to analyse how demographic information can predict the safety performance of health care professionals. Third, they are reluctant to conduct the test-retest reliability assessment but did not mention it as a limitation. Fourth, the discussion about the comparison between the findings of this study and previous studies was not in-depth.  Fifth, the practical implications should be more concrete and feasible. There should be a separate sub-section to focus on the practical implications.

Reviewer 3 Report

The authors have addressed my comments and the manuscript can be published after minor revision. my comments are as follows:

  • no need to mention in the text about the permission you obtained from an author. BTW, you need to make sure if you need permission from the publisher of that paper as well. you may discuss this matter the editors at MDPI prior to publication.
  • the added parts in the discussion is like literature review. please discuss your findings and compare it with other research. those information that has been given in discussion should be moved to literature review section.
